# Contrasting Effects of Grazing in Shaping the Seasonal Trajectory of Foliar Fungal Endophyte Communities on Two Semiarid Grassland Species

**DOI:** 10.3390/jof9101016

**Published:** 2023-10-14

**Authors:** Xin Dong, Feifei Jiang, Dongdong Duan, Zhen Tian, Huining Liu, Yinan Zhang, Fujiang Hou, Zhibiao Nan, Tao Chen

**Affiliations:** 1State Key Laboratory of Herbage Improvement and Grassland Agro-Ecosystems, Center for Grassland Microbiome, College of Pastoral Agricultural Science and Technology, Lanzhou University, Lanzhou 730000, China; dongx21@lzu.edu.cn (X.D.); jiangff19@lzu.edu.cn (F.J.); duandd17@lzu.edu.cn (D.D.); tianzh20@lzu.edu.cn (Z.T.); liuhn21@lzu.edu.cn (H.L.); 220220901660@lzu.edu.cn (Y.Z.); cyhoufj@lzu.edu.cn (F.H.); zhibiao@lzu.edu.cn (Z.N.); 2Sichuan Zoige Alpine Wetland Ecosystem National Observation and Research Station, Institute of Qinghai-Tibetan Plateau, Southwest Minzu University, Chengdu 610041, China

**Keywords:** foliar endophytic fungal community, grassland, grazing, plant community characteristic, plant nutrient, seasonal trajectory

## Abstract

Fungal endophytes are harboured in the leaves of every individual plant host and contribute to plant health, leaf senescence, and early decomposition. In grasslands, fungal endophytes and their hosts often coexist with large herbivores. However, the influence of grazing by large herbivores on foliar fungal endophyte communities remains largely unexplored. We conducted a long-term (18 yr) grazing experiment to explore the effects of grazing on the community composition and diversity of the foliar fungal endophytes of two perennial grassland species (i.e., *Artemisia capillaris* and *Stipa bungeana*) across one growing season. Grazing significantly increased the mean fungal alpha diversity of *A. capillaris* in the early season. In contrast, grazing significantly reduced the mean fungal alpha diversity of endophytic fungi of *S. bungeana* in the late season. Grazing, growing season, and their interactions concurrently structured the community composition of the foliar fungal endophytes of both plant species. However, growing season consistently outperformed grazing and environmental factors in shaping the community composition and diversity of both plant species. Overall, our findings demonstrate that the foliar endophytic fungal community diversity and composition differed in response to grazing between *A. capillaris* and *S. bungeana* during one growing season. The focus on this difference will enhance our understanding of grazing’s impact on ecological systems and improve land management practices in grazing regions. This variation in the effects of leaf nutrients and plant community characteristics on foliar endophytic fungal community diversity and composition may have a pronounced impact on plant health and plant–fungal interactions.

## 1. Introduction

Leaves cover an area exceeding 6.4 × 10^8^ km^2^ of the Earth’s surface, making them a crucial habitat for terrestrial microbial communities [1]. Fungal endophytes have been observed in the leaves of all plant species surveyed thus far, and they are essential in the ecosystem [2,3]. The roles of fungal endophytes range from beneficial to detrimental, affecting the growth and reproduction of host plants and ecosystem functioning [4,5]. Some endophytes can enhance plant defences and increase host plant tolerance to various environmental stressors, such as drought, nutrient deficiency, temperature fluctuations, pathogen attacks, competition pressures, and herbivore damage [6,7]. However, endophytes may also lead to resource extraction, suppression of plant defences, disease development, and acceleration of leaf senescence or decomposition [8,9]. At the ecosystem level, these interactions contribute to nutrient cycling and plant community successions, driving ecosystem services [10].

Foliar fungal endophytes occur within a complex ecosystem context and are influenced by several biotic and abiotic factors across various scales, ranging from the cellular to the ecological level [11,12,13]. At the leaf scale, the infection and establishment of endophytes are affected by host resource availability [14,15], stoichiometry [16], host plant morphological traits [17], and the interplay between microbiomes [18]. At larger spatial scales, foliar fungal endophytes are also affected by factors such as climate (e.g., temperature, precipitation) [19,20], nutrient input [21], genetic or community variation in host and nonhost plants [22], spatial pattern (e.g., elevation and latitude) [23], and human activities [24]. Additionally, previous studies have identified seasonal trajectories in the diversity and composition of endophyte communities [25,26,27,28]. The potential mechanisms contributing to this phenomenon are multifaceted, encompassing temporal shifts in climate patterns, leaf characteristics, and fungal life cycles [29,30,31].

Livestock grazing is a globally pervasive and pivotal land use practice that significantly influences plant, animal, and microbial community dynamics and their intricate interrelationships [32,33,34]. The positive or negative associations observed between endophytes and herbivores suggest facilitation or antagonism resulting from one or more direct or indirect mechanisms [35,36]. Considering that plants disperse endophytes horizontally through spores or hyphal fragments, grazing potentially enhances endophyte colonization by creating wounds [37,38] or contributing to their dispersal between hosts [7]. However, some studies have indicated that removing plant leaves directly would also lead to a decrease in microbial abundance [39]. In addition, grazing can modify plant community diversity and composition, which indirectly alter foliar fungal endophyte communities [40,41,42]. Moreover, grazing has the potential to alter secondary metabolites such as plant leaf stoichiometry and defensive chemicals, which may impact the composition and diversity of foliar fungal endophytic communities [43,44]. The presence of herbivores, for instance, can influence the availability of nutrients in plants and contribute to colonization when plant tissues exhibit a higher nitrogen content [45]. Various plants employ distinct response strategies to grazing, which are associated with their growth, nutritional, or physiological traits [46,47,48,49,50]. As integral components of host plants, microorganisms inevitably exhibit diverse response strategies to grazing in accordance with their hosts. Currently, our understanding of the mechanisms and response patterns of foliar fungal endophytes under grazing during growing seasons is limited. Furthermore, there is a lack of knowledge regarding the variations in foliar fungal endophytes among different host plants.

The grasslands in Northwest China occupy vast semiarid and arid areas and are sensitive to grazing disturbance [51]. *Artemisia capillaris* Thunberg (Asteraceae) and *Stipa bungeana* Trin. (Poaceae) are two dominant perennial species with important ecological and economic values in Northwest China [32]. Previous studies have revealed that *A. capillaris* is favoured by livestock primarily during the early season, whereas as plant tissue lignification intensifies throughout the growing season, the frequency of livestock feeding declines. In contrast, livestock preferred to feed on the upper portion of *S. bungeana* at all times, and it was less tolerant of grazing [32,52]. In the present study, we aimed to investigate (ⅰ) the variation in foliar endophytic fungal community composition and diversity in response to grazing between *A. capillaris* and *S. bungeana* across the full growing season; and (ⅱ) the factors driving the community composition and diversity of foliar fungal communities of *A. capillaris* and *S. bungeana.* To this end, we conducted a long-term (18 yr) grazing experiment and sampled the foliar fungal communities of *A. capillaris* and *S. bungeana* in the early, middle, and late growing seasons from control and grazed treatments. We expected the following outcomes: (i) grazing will decrease the mean fungal alpha diversity of *A. capillaris* in the early season, decrease the mean fungal alpha diversity of *S. bungeana* in the middle and late seasons, and change the community composition; (ii) growing season and grazing × growing season interactions will significantly affect foliar fungal community diversity and composition; and (iii) leaf nutrients, plant community characteristics, and defensive chemicals would be significantly correlated with the foliar endophytic fungal community. By performing this research, we expected to establish a theoretical foundation for understanding plant–herbivore–microbe interactions in grazed ecosystems subjected to increasing anthropogenic intervention.

## 2. Results

The seasonal trajectory of foliar fungal diversity exhibited divergent patterns between the grazed and control treatments. Mean fungal endophytic diversity tended to decrease by the middle season before increasing by the end of the growing season (Figure 1; Table 1). The mean fungal endophytic diversity of *A. capillaris* and *S. bungeana* showed contrasting responses to livestock grazing across the growing season (Figure 1; Table 1). The fungal diversity of *A. capillaris* was significantly higher in the grazed treatment than in the control treatment in the early growing season, whereas the fungal diversity of *A. capillaris* did not differ significantly between the grazed and control treatments at the middle and end of the growing season (Figure 1B; Table 1; F = 24.20; *p* < 0.001 for the interaction of grazing treatment and growing season). In contrast, in the early and middle growing seasons, the fungal diversity of *S. bungeana* did not differ significantly between the grazed and control treatments, whereas the fungal diversity of *S. bungeana* was lower in the grazed treatment than in the control treatment towards the end of the growing season (Figure 1A; Table 1; F = 10.91; *p* = 0.004 for the interaction of grazing treatment and growing season).

The composition of the foliar fungal community shifted during the growing season and differed between the grazed and control treatments. According to PCoA, at the ASV level, grazing, growing season, and their interaction significantly affected the community composition of fungal endophytes (Figure 2A,C; Table 1). For *A. capillaris*, there was a greater variation between the three seasons, and grazing significantly affected fungal community composition in all three seasons (Figure 2A; Table 1), with season explaining the largest variation (44.6%) in community composition and the interaction and grazing explained 11.8% and 9.4% of the variation in community composition, respectively (Table 1). The interaction between season and grazing explains 43.8% of the variation in community composition, and season and grazing explains 18.3% and 5.8% of the variation in community composition, respectively, for *S. bungeana* (Table 1). Strong grazing effects were observed for some fungal classes (Figure 2B,D; Table 1). The grazing-induced changes in fungal community composition for *A. capillaris* were due to the decrease in the relative abundance of *Tremellomycetes* in the early season, the increase in the relative abundance of *Tremellomycetes*, *Eurotiomycetes*, and *Cystobasidiomycetes* in the middle season, and the decrease in *Leotiomycetes* in the late season (Figure 2B). For *S. bungeana*, grazing-induced changes in fungal community composition were due to the decrease in the relative abundance of *Dothideomycetes* in the middle season and to the decrease in *Cystobasidiomycetes* and the increase in *Saccharomycetes* in the late season (Figure 2D). The shift in community composition during the growing season was mainly a result of species turnover and only to a minor extent a result of species gain (βJTU ≫ βJNE; Appendix A). Similarly, the differences in community composition between the grazed and control treatments were mainly attributed to species turnover (βJTU ≫ βJNE; Appendix A).

We screened for influencing factors by fitting linear regressions. In the case of *A. capillaris*, fungal community diversity was found to be correlated with plant Shannon diversity, biomass, nitrogen-phosphorus ratio (N:P ratio), total flavonoids, and tannins, while plant richness, abundance, cover, and total flavonoids, and total phosphorus was correlated with fungal community diversity for *S. bungeana* (Table 2). The multimodel inference and model averaging showed that the most important moderators were the growing season and grazing for *A. capillaris* and the growing season for *S. bungeana* (Figure 3; Appendix A). In addition to the experimental factors, the second most important moderators for both focal hosts were plant community characteristics, including plant Shannon diversity and biomass for *A. capillaris*, and plant richness, abundance, and cover for *S. bungeana.* For *S. bungeana*, plant community characteristics were more important than grazing (Figure 3; Appendix A). Except for the plant Shannon diversity for *A. capillaris*, the mean fungal endophytic diversity had a negative relationship with all other plant community characteristics (*p* < 0.05; Appendix A; Appendix A). Additionally, leaf nutrients and defensive chemicals also influenced the endophytic fungal community diversity but with less importance (Figure 3; Appendix A). Linear regression analysis indicated that total flavonoids, tannins, and the N:P ratio had a significant positive relationship with *A. capillaris*; conversely, total flavonoids had a negative relationship and total phosphorus had a positive relationship with *S. bungeana* (Appendix A; Appendix A).

Similar to fungal community diversity, the most important factor influencing community composition was growing season, and the second most important factor was grazing for both focal hosts. In addition to these two experimental factors, there were significant effects of leaf nutrients but not plant community characteristics on fungal community composition. Total flavonoids also significantly impacted the fungal community composition of *S. bungeana* (Figure 4; Appendix A).

## 3. Discussion

In this study, we investigated the contrasting effects of grazing in shaping the seasonal trajectory of foliar fungal endophyte communities of *Artemisia capillaris* and *Stipa bungeana*. Grazing significantly increased the community diversity of *A. capillaris* in the early season but significantly reduced the diversity of the foliar endophytic fungal community of *S. bungeana* in the late season. Both grazing and season strongly affected fungal community composition. Overall, season was more critical than grazing in shaping the composition of the foliar fungal community. Additionally, we detected a strong interactive effect of season and grazing on the community composition of *A. capillaris* and the community diversity and composition of *S. bungeana*. When partitioning the total β diversity into its components of turnover and changes in species richness, we found that the differences in the fungal community composition among the early, middle, and late seasons and between the grazed and control treatments were mainly explained by species turnover.

Grazing, season, and their interactions also affected aspects of local plant community characteristics, leaf nutrients, and defensive chemicals that were expected to affect foliar endophytic fungal diversity within our two focal hosts. Although high seasonal variation dominated the diversity and composition of endophytic fungal communities, grazing and other factors also affected the endophytic fungal communities. In addition to the experimental factors, plant community characteristics had the greatest impact on fungal community diversity and were negatively correlated with ENSPIE (except for the plant Shannon index in *A. capillaris*). In contrast to fungal community composition, plant community characteristics had less impact, and leaf nutrients were more nutritious.

### 3.1. Impact of Grazing in Shaping the Seasonal Trajectory of the Foliar Fungal Community

Our observation that seasonal variation had a greater effect on the foliar fungal community is in line with several previous studies [19,53,54]. The strong seasonal changes in the foliar fungal community might be linked to physical and chemical changes in leaves and variations in environmental conditions [27,55]. In this study, the endophytic fungal community diversity of both *A. capillaris* and *S. bungeana* tended to decrease by the middle season before increasing by the end of the growing season. We also detected a shift in community composition, both of which were inextricably linked to the climatic conditions of the region. The region where we established the experiment is characterised by high summer temperatures, extreme drought, low rainfall, and a rainy autumn [32]. Many previous studies have shown that warming and drought decrease endophytic microbial community activity and diversity and alter community composition [19,56]. We also detected that the shift in community composition was mainly the result of species turnover rather than the acquisition of new species [19,57].

Grazing plays a significant role in shaping the seasonal trajectory of the endophytic fungal communities of the two host plants. Our experimental grazing increased the mean fungal endophytic diversity of *A. capillaris* in the early season but decreased that of *S. bungeana* in the late season, which may be related to variations in the growth–defence strategy among different plants and the feeding nature of herbivores. Previous studies have found that *A. capillaris* is preferred by livestock only in the early season, and as plant tissue lignification increases over the growing season, the livestock feeding frequency decreases. Thus, grazing in the early season may have promoted the entry of endophytic fungal spores into leaf tissue and increased the diversity of endophytic fungi due to the rapid growth of *A. capillaris* during this period, when the removal of microbial survival space and resources by livestock was not significant. In contrast to *A. capillaris,* we found that grazing significantly decreased the foliar endophytic fungal diversity of *S. bungeana* during the late growing season. In the early and middle growing seasons, the abundant nutrients of *S. bungeana* leaves provide resources for endophytic fungal growth; thus, the fungal diversity is high, but as the growth period progresses, *S. bungeana* may allocate more energy towards compensatory growth to enhance tolerance against grazing stress imposed by livestock, consequently leading to a reduction in endophytic fungal diversity. In terms of livestock feeding habits, sheep prefer to eat the upper part of *S. bungeana* leaves at all times of the growing season. With continuous grazing, the reduction in *S. bungeana* leaves leads to a reduction in photosynthetic products and reduces the survival space and resources of endophytic fungi, leading to a decrease in endophytic fungal diversity by grazing in the late growing season. In addition, a previous study on fungal diseases of *S. bungeana* leaves showed that grazing increased the severity of *S. bungeana* leaf spot disease, while pathogenic fungi in the leaves competed with endophytic fungi for resources, thus inhibiting the growth of endophytic fungi.

Another study on the effect of grazing on *Leymus chinensis* leaf microorganisms reported a decrease in fungal α diversity under a moderate grazing treatment, and this consistency with *S. bungeana* may represent the pattern’s possible universality [53]. Interestingly, in a previous study regarding the response of root endophytic fungi to grazing, it was found that moderate grazing increased root fungal diversity [58]. In addition, grazing did not significantly affect endophytic fungal communities in plant stems in another study [53]. These different responses may indicate a spatial ecological niche difference in plant endophytes under grazing.

We further found that grazing and its interaction with growing season influenced the fungal community composition. In line with this finding, several observational studies have reported shifts in the composition of the microbial community along a grazing intensity gradient and between different grazing methods, which are often used as an experimental design to investigate the response mechanisms of the microbiome to grazing [59,60,61]. However, most studies have focused on the effects of grazing on the soil and root microbiome. At this stage, it is unclear whether differences among studies related to plant endophytic fungi are caused by different grazing intensities, herbivores, strategies, or locations. A promising avenue for future research would be to conduct comprehensive experimental studies involving multiple perspectives (grazing intensity, herbivore, strategy, and location), thereby exploring the response mechanism of plant microbes under different grazing conditions.

### 3.2. Relationships between Foliar Fungal Community and Plant Community Characteristics, Leaf Nutrients and Defensive Chemicals

The foliar fungal community may be affected by season and grazing, and it could also be affected by plant communities, nutrients, and defensive chemicals [62,63,64]. With the results of the model-averaged analysis, we found that the effects mediated by changes in the plant community were apparently consistent, and that plant community characteristics were the most important factors in addition to experimental factors, influencing the diversity of foliar endophytic fungal communities for both focal host plants. Notably, fungal diversity declined with plant community characteristics. This strong coupling between plant communities and microorganisms has been confirmed by previous studies [65,66,67,68] and confirms the need to enhance the understanding of this strong correlation between plants, microbes, and ecosystems in ecological research.

The increase in fungal diversity with decreasing plant community characteristics is contrary to island biogeography theory, one of the main theories in ecology. It has been demonstrated that island biogeography theory can be applied to explain the microbial-host relationship [69,70], where plant hosts act as islands, and it can be predicted that an increase in host community diversity would lead to an increase in microbial diversity [71,72]. However, island biogeography theory assumes that dominant and inferior species have the same fitness, and that the dynamics of richness depend strictly on identical species gains or losses [73]. In contrast, we included fitness differences and competitive hierarchies; after excluding the less competitive species, the increase in dispersal can instead cause a decrease in diversity [14]. In our study, the increased plant community characteristics led to a more rapid spread of competitively dominant fungi, leading to a decline in fungal diversity within a host individual; notably, diversity and evenness were highly correlated with ENSPIE. Furthermore, host plants may devote more resources to their own growth and reproduction or to other microbes, meaning that endophytic fungi are at a competitive disadvantage [74]. It is also possible that changing plant community characteristics have altered the local microenvironment in a direction more favourable to smaller fungal taxa [75]. In the future, more research is needed to reveal the potential mechanisms.

We also analysed the effects of various factors on the endophytic fungal community composition. In contrast to community diversity, plant community characteristics barely affected composition, while plant nutrients explained the most variation. Similarly, the effects of plant nitrogen, phosphorus, and carbon on the composition of endophytic fungal communities were demonstrated in previous studies by Meng, Wu, and Yang [76,77,78]. In addition, we did not find evidence for the pattern of defensive chemicals affecting the endophytic fungal community, although we detected weak effects of total flavonoids and tannins. However, a study on *Pinus monticola* showed that herbivores changed the leaf endophytic fungal community because herbivores induced defensive chemicals in the plant and destroyed the protective structures of the sapling (e.g., bark, leaf cuticle), independent of the nutritional status of the leaves [15]. Moreover, in our study, we found that the direction and strength of the effects of defensive chemicals on fungal endophyte diversity and composition with hosts differed greatly between *A. capillaris* and *S. bungeana*. We therefore predict that host species influence the effect of defensive chemicals on endophytic fungal communities, but at this stage, it is unclear whether these differences among studies are a result of differences in the experimental setup or the spatial scale.

We note that although we detected many effects on endophytic fungal diversity and composition, direct manipulation of local plant communities, nutrients, or chemical defence substances was not included in our experimental design, which makes it difficult to provide the strongest evidence about the effects of these factors on endophytic fungal communities. In addition, controlled isolation and culture experiments of endophytic fungi would contribute to our understanding of the mechanism that governs endophyte community dynamics. Furthermore, grazing experiments can be more refined, and previous studies on grazing usually focus on grazing intensity, grazing livestock types, or split grazing effects into the removal of plant shoot tissue, dung and urine return, and trampling [59,60,61]. Finally, the enormous seasonal variation in foliar fungal endophytes is consistent with other studies, suggesting the need to follow the entire cycle of host plants from seed to litter if we are to understand the processes underlying dominant microbial community colonisation and maintenance.

Although there is a wealth of data available on microbial diversity and composition, the mechanistic understanding of causal factors remains unknown. The large number of observational studies provides a solid foundation and an important starting point for subsequent studies, which should focus more on identifying underlying drivers through controlled experimentation and making predictive observations. For example, grazing experiments have revealed that grazing can affect plant symbiont communities in several ways, including direct removal, changing local plant communities, promoting microbial horizontal spread, and changing plant metabolites. However, the large span of seasonal grazing experiments is correlated with many other factors (e.g., climate, micro food network, local plant community, host species, and herbivore diet) that also change microbial communities. For this reason, controlled experiments that are replicated at the temporal scale should be integrated with existing community ecological theoretical frameworks, through which we can understand the mechanisms that determine microbial communities.

## 4. Materials and Methods

### 4.1. Study System and Experimental Design

This study was conducted within an existing long-term grazing experiment, which was initiated in 2001 at the Tian Shui Grassland Research Station of Lanzhou University, situated in Huan County, Gansu Province, People’s Republic of China (37°12′ N, 106°82′ E, 1650 m a.s.l.). This region experiences a typical semiarid monsoon climate characterised by an average annual temperature of approximately 7.1 °C and annual rainfall averaging approximately 360 mm (>80% occurring from June to September) [79]. The soil type prevalent in this area is classified as Cambisol [32]. Grassland plants turn green in mid-April, experience peak growth from late June to late August, and senesce in late September. The vegetation of the grassland represents a typical temperate steppe ecosystem dominated by the forb *Artemisia capillaris* Thunberg (Asteraceae), the semishrub *Lespedeza davurica* (Laxm.) Schindl (Fabaceae), and bunchgrass *Stipa bungeana* Trin. (Poaceae) [79].

The grazing experiment was conducted using a completely randomised design, incorporating four levels of grazing intensity (0, 2.7, 5.3 and 8.7 sheep/ha); each grazing intensity consisted of three replicated plots, each with an area of 0.5 ha per plot. The primary utilisation of local grasslands is sheep grazing, with most of the grasslands experiencing moderate levels of grazing [52]. Therefore, we chose a control treatment (0 sheep/ha) and a medium-intensity sheep-grazed treatment (5.3 sheep/ha) as our study sites. The sample plots of the grazing experiment encompassed a total area of 1.5 ha and were inhabited by a collective count of 8 sheep. The grazing plots were rotationally grazed three times (10 days each time, with a rotation interval of 30 days) from early June to early September (Appendix A).

*A. capillaris* and *S. bungeana* thrive in a diverse range of habitats across Northwestern China [80,81]. Similar to other plant species, the leaves of *A. capillaris* and *S. bungeana* harbour a rich fungal community comprising both epiphytic and endophytic fungi [52,82,83]. These fungi were widely distributed throughout our experimental sites, exhibiting distinct characteristics and responses to grazing [52]. *S. bungeana* exhibits moderate palatability, is preferred by livestock, and has great resilience and resistance to grazing [84]. In contrast, *A. capillaris* is primarily consumed by livestock during the early growth stage only and has poor grazing resistance.

### 4.2. Plant Sampling

We collected leaf samples from the control and grazed treatments at three time points: the beginning (10 June 2019, after the first grazing), middle (10 August 2019, after the second grazing), and end (8 October 2019, after the third grazing) of the growing season. Both the grazed treatment and the control treatment consisted of three replicated plots, each with an area of 0.5 ha per plot. In each replicated plot, all fresh leaves of eight equal lengths of *A. capillaris* and *S. bungeana* plants without visible disease or injury were randomly and uniformly collected as samples from a Z-shaped transect in each plot, and all leaves from the grazed and control treatments were mixed separately and then divided into six replicates. The leaves were stored in a cooler at approximately 4 °C until they underwent surface sterilisation in the laboratory. This involved immersing them in water for 30 s, followed by a 2-min immersion in 75% ethanol, then a 5-min immersion in a solution of 2.5% NaClO (with 0.1% Tween 80), and finally rinsing with sterile distilled water three times. After sterilisation, a portion of the samples was stored at −80 °C, and 5 g of sample per replicate was utilised for sequencing purposes, while the remaining leaf samples were subjected to baking at 75 °C until reaching constant weight. Subsequently, they were ground into fine powder using a bead mill (Retsch MM400, Retsch, Haan, Germany), 10 g of sample per replicate was utilised for measuring nutrients, and 5 g of sample per replicate was utilised for determination of chemical defence substances.

Furthermore, plant abundance, cover, biomass, and richness were sampled at the start (10 June 2019), middle (10 August 2019), and end (8 October 2019) of the growing season in each plot. Briefly, we randomly established six 1 × 1 m quadrats along a W-shaped transect and recorded the number of *A. capillaris* and *S. bungeana* individuals (quantified based on the number of stems/ramets) the percent cover of each species (visually estimating the percent cover to the nearest 1% of each plant species), and richness (total number of species) per quadrat. For biomass, we clipped all plant biomass in each quadrat, dried the biomass at 60 °C to a constant mass and weighed the dried samples to the nearest 0.01 g. Additionally, we calculated the ‘Shannon index’ for each sample.

Leaf nitrogen and phosphorus were determined by a Smartchem450 Analyser (AMS, Florence, Italy), and the concentration of leaf carbon was determined by ferrous sulfate titration. The determination of three defensive chemicals (tannins, total phenols, and total flavonoids) was performed by Shanghai Enzyme-linked Biotechnology Co., Ltd. (Shanghai, China).

### 4.3. Molecular Methods and Bioinformatics

Leaves were ground in liquid nitrogen with a mortar and pestle, and total genomic DNA was extracted using the FastDNA^®^ Spin Kit (MP Biomedicals, Santa Ana, CA, USA) and standardised to 20 ng/L. Next, we generated fungal genomic amplicon libraries by amplifying fungal samples using the internal transcribed spacer (ITS) region [85,86]. Each sample was barcoded with unique 7 base-pair (bp) sequences, and we used the forwards primer ITS7F (-GTGARTCATCGAATCTTTG-) [87] and reverse primer ITS4 (-TCCTCCGCTTATTGATATGC-) [88]. PCRs for each sample were carried out in triplicate 20-μL reactions with 0.8 μL of each primer at 5 μmol·L^−1^, 10 ng template DNA, 2 μL dNTPs at 2.5 mmol·L^−1^, 0.4 μL FastPfu Polymerase (TransGen AP221-02; TransStart FastPfu DNA Polymerase; TransGen Biotech, Beijing, China), 4 μL 5× FastPfu buffer, 0.2 μL BSA, and certified DNA-free PCR water, according to the following procedures: 95 °C for 2 min, 32 cycles of 95 °C for 30 s, 53 °C for 30 s and 72 °C for 30 s, and 72 °C for 5 min. PCR amplifications were all performed on the ABI GeneAmp 9700 PCR system (Applied Biosystems, Waltham, MA, USA). Then, replicated amplicons were pooled and visualised on 2% agarose gels using SYBR Safe DNA gel stain in 0.5× TBE. Subsequently, amplicons were cleaned using the AxyPrep DNA Gel Extraction Kit (Axygen Biosciences, Union City, CA, USA), quantified by PicoGreen dsDNA Quantitation Reagent and QuantiFluor-ST Fluorometer (Promega Corp., Madison, WI, USA), and combined with equimolar ratios into a single tube. The barcoded pyrosequencing for fungi was performed on an Illumina MiSeq PE300 platform (Illumina, San Diego, CA, USA) at Shanghai Majorbio Bio-Pharm Technology Co., Ltd., Shanghai, China.

Sequence data from the MiSeq runs were combined and analysed using an analysis pipeline adapted from Liu et al. (2021) [89]. First, raw amplicon paired-end reads were grouped based on their barcode sequences (demultiplexing). Subsequently, the paired reads were merged to obtain amplicon sequences, and barcodes and primers were removed. Furthermore, a quality-control step was used to remove low-quality amplicon sequences, and we filtered sequences by removing short sequence reads and homopolymers using Mothur v.1.34.4 [90]. All these steps were completed using USEARCH v. 11.0.1 [91]. The clean sequences were then diagnosed as amplicon sequence variants (ASVs) at a 97% cut-off using -unoise3 in USEARCH v. 11.0.1 [92] with chimeric sequences removed. A feature table (ASV table) was obtained by quantifying the frequency of the feature sequences in each sample, and then we removed plastids and non-fungi adopting a confidence threshold of 0.8. Finally, equal resampling was performed using the vegan package to normalise ASVs by subsample.

We obtained a total of 2194428 DNA sequences from the 36 leaf samples of *A. capillaris* (40,067–68,805 reads per sample; mean = 609,560.3) that passed quality filtering and detected 681 unique fungal ASVs. Note that all ASVs identified as fungi (matched and unmatched to a lower taxonomic level) were used in subsequent analyses. These ASVs were mostly from 4 phyla, 20 classes, 47 orders, and 137 genera. OTU frequencies by phylum were as follows: Ascomycota (56.6%), Basidiomycota (24.4%), Glomeromycota (1.3%), and Chytridiomycota (0.4%). For *S. bungeana*, we obtained a total of 2,151,730 DNA sequences from the 36 leaf samples of *S. bungeana* (41,562–68,324 reads per sample; mean = 59,770.3) that passed quality filtering and detected 548 unique fungal ASVs. Note that all ASVs identified as fungi (matched and unmatched to a lower taxonomic level) were used in subsequent analyses. These ASVs were mostly from 3 phyla, 20 classes, 49 orders, and 128 genera. OTU frequencies by phylum were as follows: Ascomycota (56%), Basidiomycota (29.9%), and Chytridiomycota (0.3%).

### 4.4. Statistical Analysis

All analyses were conducted in R v.4.1.2 (R Core Team, 2021). In assessing fungal diversity, we used the effective number of species based on the probability of interspecific encounter (ENSPIE), a measure of diversity that is more robust to the effects of sampling scale and less sensitive to the presence of rare species than species richness [93,94,95]. ENSPIE estimates the number of equally abundant species. We calculated ENSPIE as 1 = PS i¼1 p2i, where S is the total number of species and pi is the proportion of the community represented by species I [95]. Across all data, ENSPIE was positively correlated with evenness (r*_Stipa bungeana_* = 0.85, r*_Artemisia capillaris_* = 0.97) and the Shannon index (r*_Stipa bungeana_* = 0.89, r*_Artemisia capillaris_* = 0.97).

To investigate the impact of grazing and season, we modelled ENSPIE as a function of the fixed effects ‘grazing, ‘season’, and their interactions using linear mixed effects models with the function ‘lmer’ in the LME4 package. To account for repeated sampling of the same treatment, we included the random effect ‘replication’. To test for significance, we used the function ‘ANOVA’ in the CAR package [96]. For each significant fixed effect, we calculated the marginal R^2^ using the function ‘r.squaredGLMM’ in the MUMIN package [97].

When we incorporated all factors (plant community characteristics, plant nutrients, and defensive chemicals) into our models, we used multimodel inference and model averaging. To screen key factors, we initially conducted individual linear regression analyses with each factor as the independent variable and ENSPIE as the dependent variable. Only those factors that exhibited a significant correlation with ENSPIE (*p* < 0.05) were selected for further analysis. Then, we fitted a mixed effects model as the base model for the model averaging. In this model, experimental treatments, seasons, plant community characteristics, nutrients, and defensive chemicals were treated as fixed effects. We used the dredge function in the MuMIn library to fit all possible models and the model.avg function to estimate parameter values, errors, and the Akaike information criterion (AICc)-weighted importance for all models within four AICc units of the top model [98]. In addition, we selected the correlation coefficient according to the *p* value of the significance test (retaining only correlation coefficient with *p* < 0.05) and the correlation r-value (that is, screening according to the correlation strength and only retaining the correlation coefficient whose absolute value was greater than or equal to 0.5) to perform the correlation analysis. Finally, to investigate the impact of grazing, season, and their interactions on these factors, we fitted each factor as a function of the fixed effects ‘grazing, ‘season’, and their interactions using linear mixed effects models.

To assess the drivers of the fungal community composition, we first modelled multivariate fungal community composition as a function of grazing, season and their interactions using PERMANOVA as implemented in the function ‘adonis2’ (with the argument by = margin) in the VEGAN package [99]. We used principal coordinate analysis (PCoA) to detect the variations in community composition among the three growing seasons and between the grazed and control treatments within each growing season. Then, we added other factors into our model, and we modelled multivariate fungal community composition as a function of experimental treatments, plant community characteristics, nutrients, defensive chemicals, and their interactions using PERMANOVA. All models were run using Bray–Curtis dissimilarity metrics. For each factor, we extracted partial-R^2^ values from the PERMANOVA model output [99]. Because we used marginal models, the value associated with the partial R^2^ of interaction terms accounted only for the contribution of these interactions, given that all other explanatory variables (including major effects and their interactions) remain constant [100]. For this reason, we needed to interpret the partial R^2^ associated with these interactions as their independent contribution to the model.

Because PERMANOVA does not separate the effects of species turnover from changes in species richness, we disentangled these responses using the functions ‘beta.temp’ and ‘beta.pair’ from the package BETAPART [101]. We first analysed the changes in community composition during the growing season by computing dissimilarity values between the early and middle seasons and between the middle and late seasons. We then disentangled responses to grazing. In both analyses, we partitioned the total β diversity (Jaccard dissimilarity calculated based on presence–absence data) into two indices, where βJTU was the turnover component of Jaccard dissimilarity and βJNE was the species gain or loss component of Jaccard dissimilarity.

## Figures and Tables

**Figure 1 jof-09-01016-f001:**
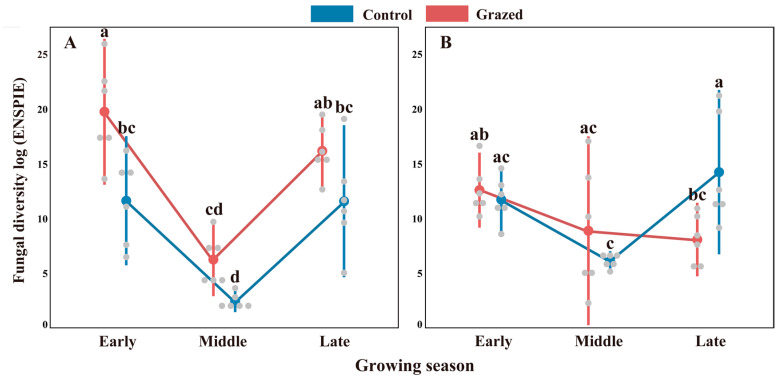
Effects of grazing treatment on foliar fungal endophyte diversity of *Artemisia capillaris* (**A**) and *Stipa bungeana* (**B**) in the early, middle, and late growing seasons. The large circles represent the mean values, and the error bars represent standard errors. The small circles represent raw data points (*n* = 6). Within each panel, different lowercase letters indicate significant differences between treatments at *p* < 0.05 (Tukey’s HSD).

**Figure 2 jof-09-01016-f002:**
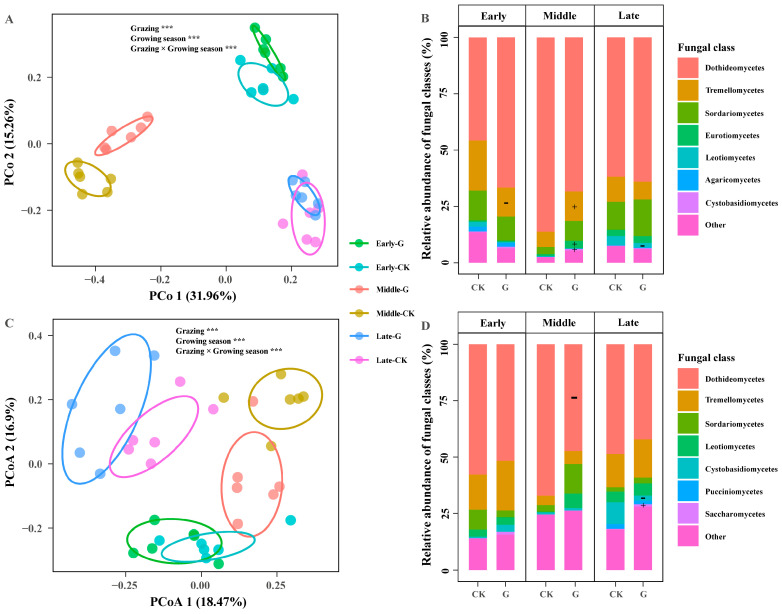
Responses of community composition of foliar fungal endophyte of *Artemisia capillaris* (**A**,**B**) and *Stipa bungeana* (**C**,**D**) to grazing treatment during growing season (early, middle, and late). G, grazed; CK, control. Visualization is based on principal coordinate analysis using Bray–Curtis metrics at the ASV level. The effects of grazing treatment, growing season, and their interactions on fungal community composition were assessed using permutational multivariate analysis of variance (***, *p* < 0.001). Stacked bar plots indicate the relative abundance (%) of foliar fungal classes of *Artemisia capillaris* (**B**) and *Stipa bungeana* (**D**) as affected by grazing treatment during growing season. The symbols ‘+’ and ‘−’ indicate a significant (*p* < 0.05) increase and decrease, respectively, in response to grazing based on a linear mixed-effect model.

**Figure 3 jof-09-01016-f003:**
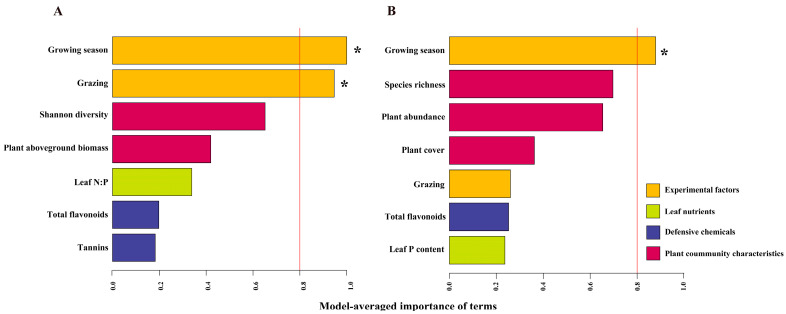
Variable importance of experimental factors, leaf nutrients, defensive chemicals, and plant community characteristics for the ENSPIE of (**A**) *Artemisia capillaris* and (**B**) *Stipa bungeana*. The importance values are derived from a model-averaged analysis including variables as moderators in the model. The asterisks (*) indicate significant (*p* < 0.05) moderators. Red line indicates the variable importance of 0.8.

**Figure 4 jof-09-01016-f004:**
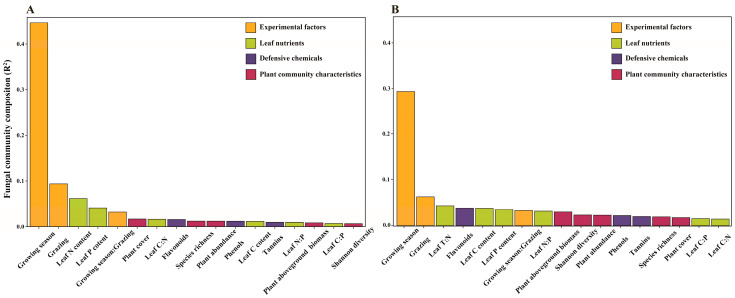
Amount of the variation (R^2^) in fungal endophyte communities in leaves of (**A**) *Artemisia capillaris* and (**B**) *Stipa bungeana* accounted for by experimental factors, leaf nutrients, defensive chemicals, and plant community characteristics. Results are from a PERMANOVA using Bray–Curtis distance and 999 permutations of the data. R^2^ is ranked in descending order.

**Table 1 jof-09-01016-t001:** Summary of the results for grazing treatment (G), growing season (S), and their interactions on the foliar fungal community of *Artemisia capillaris* and *Stipa bungeana* as modelled using linear mixed models for diversity (ENSPIE) and PERMANOVA for community composition (absolute count data). Data are represented as *p*-values and R^2^. Parameters in bold are significantly different from zero in the models in which they are included. Test statistics and degrees of freedom are reported in Appendix A, Appendix A.

Sources	Grazing (G)		Season (S)		G × S	
	*p*	R^2^	*p*	R^2^	*p*	R^2^
*Artemisia capillaris*						
ENSPIE	**<0.001**	0.174	**<0.001**	0.566	0.243	0.752
Community composition	**0.001**	0.094	**0.001**	0.446	**0.001**	0.118
*Stipa bungeana*						
ENSPIE	0.468	0.010	**0.003**	0.213	**0.004**	0.400
Community composition	**0.001**	0.058	**0.001**	0.272	**0.001**	0.091

**Table 2 jof-09-01016-t002:** Results of linear regression models for fungal endophyte diversity (ENSPIE) of *Artemisia capillaris* and *Stipa bungeana* against plant community characteristics (abundance, cover, aboveground biomass, species richness, and Shannon diversity), leaf nutrients (C, N, P, C:N, C:P, and N:P), and defensive chemicals (tannins, total flavonoids, and total phenols). Parameters in bold are significantly different from zero in the models in which they are included.

Predictors	*Artemisia capillaris*			*Stipa bungeana*		
	Estimate	R^2^	*p*	Estimate	R^2^	*p*
Plant abundance	−0.015	0.025	0.355	−0.120	0.144	**0.022**
Plant cover/log	−2.937	0.090	0.075	−0.227	0.112	**0.047**
Plant aboveground biomass	−0.164	0.299	**<0.001**	−0.305	0.012	0.518
Species richness	−0.652	0.026	0.343	−1.199	0.206	**0.005**
Shannon diversity	8.273	0.120	**0.038**	−2.795	0.033	0.292
Leaf C content	0.002	0.001	0.942	0.009	0.013	0.507
Leaf N content	0.242	0.062	0.143	0.024	0.001	0.839
Leaf P content	−4.241	0.064	0.135	5.066	0.150	**0.019**
Leaf C:N	−0.227	0.082	0.090	−0.051	0.006	0.653
Leaf C:P	0.020	0.044	0.222	−0.021	0.090	0.075
Leaf N:P	0.573	0.221	**0.004**	−0.135	0.020	0.416
Tannins	10.398	0.121	**0.038**	0.894	0.002	0.806
Total flavonoids	4.517	0.334	**<0.001**	−2.267	0.104	**0.055**
Total phenols	2.008	0.059	0.153	−1.009	0.023	0.377

The Plant cover of *Artemisia capillaris* was log-transformed to achieve normality of the residuals.

## Data Availability

All applicable data are published and referenced in the paper.

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
