# Peer review of "Contrasting Effects of Grazing in Shaping the Seasonal Trajectory of Foliar Fungal Endophyte Communities on Two Semiarid Grassland Species"

_jof, 2023, doi:10.3390/jof9101016_

Round 1

Reviewer 1 Report

In the manuscript “Contrasting Effects of Grazing in Shaping the Seasonal Trajectory of Foliar Fungal Endophyte Communities on Two Semiarid Grassland Species” the authors conducted a longterm (18 yr) grazing experiment to explore the effects of grazing on the community composition and diversity of foliar fungal endophytes of two perennial grassland species (i.e., Artemisia capillaris and Stipa bungeana) across one growing seasone. This manuscript is well organized, and the drawn conclusions are coherent with the obtained results. The paper was well written!

Lines 41 – 42: Please, arrange the keywords alphabetically.

Lines 69 – 71: I think that you should these important references as examples to support your sentence: “Livestock grazing is a globally pervasive and pivotal land use practice that significantly influences plant, animal, and microbial community dynamics and their intricate interrelationships”. I would like to suggest:

Bosso, L., et al., (2017). Plant pathogens but not antagonists change in soil fungal communities across a land abandonment gradient in a Mediterranean landscape. Acta Oecologica, 78, 1-6.

Wu, J., et al., (2021). Livestock exclusion reduces the spillover effects of pastoral agriculture on soil bacterial communities in adjacent forest fragments. Environmental Microbiology, 23(6), 2919-2936.

Lines 84 – 99 : Please, reduce this part of the manuscript.

Lines 100 – 113: Please, explain in detail you hypothesis and predictions.

Lines 83 – 101: Did you analyse your data for spatial auto-correlation?

Line 129: To delete the brackets from the letter (A) and (B) in the figure.

Line 171: To delete the brackets from the letter (A), (B), (C) and (D) in the figure.

Line 179: To delete the brackets from the letter (A), (B), (C) and (D) in the figure.

Line 183: To delete the brackets from the letter (A), (B), (C), (D), (E) and (F) in the figure.

Line 210: To delete the brackets from the letter (A) and (B) in the figure.

Line 215: To delete the brackets from the letter (A), (B), (C), (D) and (E) in the figure.

Line 221: To delete the brackets from the letter (A), (B), (C), (D) and (E) in the figure.

Line 234: To delete the brackets from the letter (A) and (B) and the grey background in the figure.

Lines 457 – 458: I think that you should these important references as examples to support your method: “we generated fungal genomic amplicon libraries by amplifying fungal samples using the internal transcribed spacer (ITS) region”. I would like to suggest:

Hechmi, N., et al., (2016). Depletion of pentachlorophenol in soil microcosms with Byssochlamys nivea and Scopulariopsis brumptii as detoxification agents. Chemosphere, 165, 547-554.

Purahong, W., et al., (2019). Characterization of the Castanopsis carlesii deadwood mycobiome by Pacbio sequencing of the full-length fungal nuclear ribosomal internal transcribed spacer (ITS). Frontiers in Microbiology, 10, 983.

Moderate editing of English language required

Author Response

Dear reviewer,

Thank you very much for giving us the opportunity to revise our manuscript. We thank the reviewers for their thoughtful comments, which have helped us to further improve the manuscript. We have given full consideration of all points and were able to incorporate the reviewers’ suggestions. Please, see further our detailed responses to the reviewers’ comments below.

Comments from Reviewer 1

In the manuscript “Contrasting Effects of Grazing in Shaping the Seasonal Trajectory of Foliar Fungal Endophyte Communities on Two Semiarid Grassland Species” the authors conducted a long term (18 yr) grazing experiment to explore the effects of grazing on the community composition and diversity of foliar fungal endophytes of two perennial grassland species (i.e., Artemisia capillaris and Stipa bungeana) across one growing season. This manuscript is well organized, and the drawn conclusions are coherent with the obtained results. The paper was well written!

Response: Thank you for your positive feedback and useful suggestions. Please, see further our detailed responses to your questions and suggestions below.

Q1. Lines 41 – 42: Please, arrange the keywords alphabetically.

Response: Thank you for your reminder. We have arranged the keywords alphabetically (Lines 43-44).

Q2. Lines 69 – 71: I think that you should these important references as examples to support your sentence: “Livestock grazing is a globally pervasive and pivotal land use practice that significantly influences plant, animal, and microbial community dynamics and their intricate interrelationships”. I would like to suggest:

Bosso, L., et al., (2017). Plant pathogens but not antagonists change in soil fungal communities across a land abandonment gradient in a Mediterranean landscape. Acta Oecologica, 78, 1-6.

Wu, J., et al., (2021). Livestock exclusion reduces the spillover effects of pastoral agriculture on soil bacterial communities in adjacent forest fragments. Environmental Microbiology, 23(6), 2919-2936.

Response: Thanks for your kind suggestions. We have included these references in the text (see references 33 and 34, Line 73 and Lines 662-667).

Q3. Lines 84 – 99: Please, reduce this part of the manuscript.

Response: Thank you for your suggestion. We have reduced this part of the manuscript and made it more condensed (Lines 86-93).

Q4. Lines 100 – 113: Please, explain in detail you hypothesis and predictions.

Response: Thank you for your valuable suggestions! We have explained our hypothesis and predictions in more detail (Lines 105-116). We have added the following:

To this end, we conducted a long-term (18 yr) grazing experiment, and sampled the foliar fungal communities of A. capillaris and S. bungeana in the early, middle and late growing seasons from control and grazed treatments. We expected that: (i) Grazing will decrease the mean fungal alpha diversity of A. capillaris in the early season, decrease the mean fungal alpha diversity of S. bungeana in the middle and late seasons and change the community composition; (ii) growing season and grazing × growing season interactions will significantly affect foliar fungal community diversity and composition; and (iii) leaf nutrients, plant community characteristics and defensive chemicals would be significantly correlated with the foliar endophytic fungal community.

Q5. Lines 83 – 101: Did you analyse your data for spatial auto-correlation?

Response: Thanks for your good question. During sampling period, we tried to avoid the differences caused by different spatial locations, so the sample points in our sampling area are set uniformly and the spatial structure of the sample site is relatively simple. The leaves were randomly and uniformly collected from eight individual A. capillaris /S. bungeana plants of equal length without visible disease or injury along a Z-shaped transect at each plot and served as the samples, all leaves from the same treatment were mixed and divided into six replicates. However, we did not obtain the geographic coordinates based on the samples, so we could not conduct spatial auto-correlation analysis.

Q6. Line 129: To delete the brackets from the letter (A) and (B) in the figure.

Response: Thank you for this useful suggestion. We have deleted the brackets from the letter (A) and (B) in the figure (Line 133).

Q7. Line 171: To delete the brackets from the letter (A), (B), (C) and (D) in the figure.

Response: We have deleted the brackets from the letter (A), (B), (C) and (D) in the figure (Line 175).

Q8. Line 179: To delete the brackets from the letter (A), (B), (C) and (D) in the figure.

Response: We have deleted the brackets from the letter (A), (B), (C) and (D) in the figure (Line 183).

Q9. Line 183: To delete the brackets from the letter (A), (B), (C), (D), (E) and (F) in the figure.

Response: We have deleted the brackets from the letter (A), (B), (C), (D), (E) and (F) in the figure (Line 187).

Q10. Line 210: To delete the brackets from the letter (A) and (B) in the figure.

Response: We have deleted the brackets from the letter (A) and (B) in the figure (Line 215).

Q11. Line 215: To delete the brackets from the letter (A), (B), (C), (D) and (E) in the figure.

Response: We have deleted the brackets from the letter (A), (B), (C), (D) and (E) in the figure (Line 220).

Q12. Line 221: To delete the brackets from the letter (A), (B), (C), (D) and (E) in the figure.

Response: We have deleted the brackets from the letter (A), (B), (C), (D) and (E) in the figure (Line 226).

Q13. Line 234: To delete the brackets from the letter (A) and (B) and the grey background in the figure.

Response: We have deleted the brackets from the letter (A) and (B) and the grey background in the figure (Line 239).

Q14. Lines 457 – 458: I think that you should these important references as examples to support your method: “we generated fungal genomic amplicon libraries by amplifying fungal samples using the internal transcribed spacer (ITS) region”. I would like to suggest:

Hechmi, N., et al., (2016). Depletion of pentachlorophenol in soil microcosms with Byssochlamys nivea and Scopulariopsis brumptii as detoxification agents. Chemosphere, 165, 547-554.

Purahong, W., et al., (2019). Characterization of the Castanopsis carlesii deadwood mycobiome by Pacbio sequencing of the full-length fungal nuclear ribosomal internal transcribed spacer (ITS). Frontiers in Microbiology, 10, 983.

Response: Thank you for this useful suggestion. We have included these references in the text (see the reference 88 and 89, Line 480 and Lines 791-796).

Q15. Moderate editing of English language required.

Response: Thanks for your suggestion. The English language of our manuscript has been further edited by the English polishing company Research Square AJE LLC.

Reviewer 2 Report

The authors intended to create a model for understanding plant-herbivore-microbe interactions in

grazing ecosystems taking into account human interventions as well.

Major issues

The authors talk about grazed grasslands, but they do not provide details about the animal species that grazed in there! So, animal species, number of animals, duration of grazing etc. are paramount details of the model and must be described.

How many leaves were collected and studied in total? Was there any strategy in selecting plants and leaves within the plants? Serious omission, please rectify.

Modelling: please create a new model by including therein the details of animals that grazed the areas from where leaves were collected.

Results. Please present the results of the new analysis (new model) additionally to the ones already included in the manuscript.

I have significant concerns about this manuscript, so the authors must be very careful when revising and must make all the changes suggested above.

Author Response

Dear reviewer,

Thank you very much for giving us the opportunity to revise our manuscript. We thank the reviewers for their thoughtful comments, which have helped us to further improve the manuscript. We have given full consideration of all points and were able to incorporate the reviewers’ suggestions. Please, see further our detailed responses to the reviewers’ comments below.

Comments from Reviewer 2

The authors intended to create a model for understanding plant-herbivore-microbe interactions in grazing ecosystems taking into account human interventions as well.

Q1. The authors talk about grazed grasslands, but they do not provide details about the animal species that grazed in there! So, animal species, number of animals, duration of grazing etc. are paramount details of the model and must be described.

Response: We feel sorry for the unclarity. The animal species, number and duration of grazing selected for our experiments were determined by the grazing utilization practices of the local grasslands. This experiment is a controlled experiment with a certain species, number and duration of grazing. The grazing animal species in our grazing experiment plots was sheep, and the grazing intensity chosen was the medium intensity (5.3 sheep/ha). The grazing experiment consisted of three replicated plots, each with an area of 0.5 ha per plot. The sample plots of the grazing experiment encompassed a total area of 1.5 ha and were inhabited by a collective count of 8 sheep. Grazing began in early June each year with 10 days of grazing and 30 days of rotational interval for a total of three grazing sessions, ending in late September for a total of three months. We have included this information into the text (Lines 421-433).

Q2. How many leaves were collected and studied in total? Was there any strategy in selecting plants and leaves within the plants? Serious omission, please rectify.

Response: We thank for the valuable comments. Both the grazed treatment and the control treatment consisted of three replicated plots, each with an area of 0.5 ha per plot. In each replicated plot, all fresh leaves of eight equal lengths of A. capillaris and S. bungeana plants without visible disease or injury were randomly and uniformly collected as samples from a Z- shaped transect in each plot, and all leaves from the grazed and control treatments were mixed separately and then divided into six replicates. After leaf sterilization, 5 g of sample per replicate was utilized for sequencing, and after drying and grinding, 10 g of sample per replicate was utilized for leaf nutrients assay, 5 g for defensive chemicals assay. We collected leaf samples from the control and grazed treatments at three time points: the beginning (10 June 2019, after the first grazing), middle (10 August 2019, after the second grazing), and end (8 October 2019, after the third grazing) of the growing season. Each treatment had six replicates, resulting in a total of 36 samples collected from A. capillaris and S. bungeana. We have added this information to the text (Lines 443-451 and Lines 455-460).

Q3. Modelling: please create a new model by including therein the details of animals that grazed the areas from where leaves were collected.

Response: Thanks for your suggestion! In order to visualize the details of the grazing treatments for this experiment, we added the following table in Supporting Information (Table S4, Lines 432-433).

Table S4. A summary of information on grazed treatment and control treatment.

Treatment

Grazed

Control

Animal species

Sheep

None

Grazing intensity

5.3 sheep/ha

0

Animal number

8 sheep

0

Total area

1.5 ha

1.5 ha

Replicated plots

3

3

Duration

Grazing began in early June with 10 days of grazing and 30 days of rotational interval for a total of three grazing sessions, ending in late September for a total of three months.

None

Q4. Results. Please present the results of the new analysis (new model) additionally to the ones already included in the manuscript.

Response: Thank you for your valuable suggestions! In our experimental design, the species and number of grazing animals and the duration of grazing were fixed. Therefore, they were not treated as variables in subsequent analyses. The purpose of our experiment was to investigate the variation in foliar endophytic fungal community composition and diversity in response to grazing between two different host plants across the full growing season. Thus, adding factors such as the species and number of grazing animals and duration of grazing, as variables in the model may deviate from our scientific hypotheses. However, this is a very meaningful suggestion that we will take into account in our future research!

Round 2

Reviewer 2 Report

No further comments. 
All my concerns have been correctly answered.